# U-shaped-aggressiveness of SARS-CoV-2: Period between initial symptoms and clinical progression to COVID-19 suspicion. A population-based cohort study

**Dan Morgenstern-Kaplan**[1]◉, **Bruno Buitano-Tang**[1]◉, **Mercedes Martínez-Gil**[1‡], **Andrea Zaldívar-Pérez Pavón**[1‡], **Juan O. Talavera**[1,2]*

**1** Centro de Investigación en Ciencias de la Salud Anáhuac (CICSA), Facultad de Ciencias de la Salud, Universidad Anáhuac México, Huixquilucan, Estado de México, México, **2** Department of Medical Education and Research, ABC Medical Center, Mexico City, CDMX, Mexico

◉ These authors contributed equally to this work.
‡ MMG and AZPP also contributed equally to this work.
* jotalaverap@abchospital.com

## Abstract

### Background

Early identification of different COVID-19 clinical presentations may depict distinct patho-physiological mechanisms and guide management strategies.

### Objective

To determine the aggressiveness of SARS-CoV-2 using symptom progression in COVID-19 patients.

### Design

Historic cohort study of Mexican patients. Data from January-April 2020 were provided by the Health Ministry.

### Setting

Population-based. Patients registered in the Epidemiologic Surveillance System in Mexico.

### Participants

Subjects who sought medical attention for clinical suspicion of COVID-19. All patients were subjected to RT-PCR testing for SARS-CoV-2.

### Measurements

We measured the Period between initial symptoms and clinical progression to COVID-19 suspicion (PISYCS) and compared it to the primary outcomes (mortality and pneumonia).

**Data Availability Statement:** Data is held in a public Data Repository at Mendeley Data DOI: http://dx.doi.org/10.17632/k6cw45366d.1.

**Funding:** The author(s) received no specific funding for this work.

**Competing interests:** The authors have declared that no competing interests exist.

## Results

65,500 patients were included. Reported fatalities and pneumonia were 2176 (3.32%), and 11568 (17.66%), respectively. According to the PISYCS, patients were distributed as follows: 14.89% in <24 hours, 43.25% between 1–3 days, 31.87% between 4–7 days and 9.97% >7 days. The distribution for mortality and pneumonia was 5.2% and 22.5% in <24 hours, 2.5% and 14% between 1–3 days, 3.6% and 19.5% between 4–7 days, 4.1% and 20.6% >7 days, respectively (p<0.001). Adjusted-risk of mortality was (OR [95% CI], p-value): <24 hours = 1.75 [1.55–1.98], p<0.001; 1–3 days = 1 (reference value); 4–7 days = 1.53 [1.37–1.70], p<0.001; >7 days = 1.67 [1.44–1.94], p<0.001. For pneumonia: <24 hours = 1.49 [1.39–1.58], p<0.001; 1–3 days = 1; 4–7 days = 1.48 [1.41–1.56], p<0.001; >7 days = 1.57 [1.46–1.69], p<0.001.

## Limitations

Using a database fed by large numbers of people carries the risk of data inaccuracy. However, this imprecision is expected to be random and data are consistent with previous studies.

## Conclusion

The PISYCS shows a U-shaped SARS-CoV-2 aggressiveness pattern. Further studies are needed to corroborate the time-related pathophysiology behind these findings.

## Introduction

Coronaviruses are single-stranded RNA organisms capable of infecting humans and other animal species [1, 2]. The most recently discovered coronavirus, Severe Acute Respiratory Syndrome Coronavirus 2 (SARS-CoV-2), is cause of the clinical entity denominated Coronavirus Disease 2019 (COVID-19). This virus initially spread in the Wuhan province in China and later to the rest of the world, causing a pandemic [3]. Reported worldwide cases are continuously growing and currently (as of July 3rd, 2020) there are over 10 million infected people confirmed and over 500,000 fatalities. Global reports reveal case-fatality rate of 4.8% and more than half of the cases are in the Americas region. In Mexico, over 230,000 cases have been reported, with over 28,000 fatalities and a case-fatality rate of 12.3%, which by far surpasses the global estimate [4].

Every human in the world is susceptible to infection, for as the mean age of infected patients is 47 years, 87% of patients lie between 30 and 79 years old. COVID-19 behaves more aggressively in older patients and in patients undergoing chronic medical conditions such as obesity, diabetes [5, 6], hypertension and other cardiovascular diseases, increasing the risk of mortality in these populations [7, 8]. Approximately 80% of cases are asymptomatic with a mild disease course, while the other 20% can be accompanied of severe complications such as pneumonia, Acute Respiratory Distress Syndrome (ARDS) and other secondary infections. Among these severe cases, 80% correspond to people over 60 years. Many of these cases can be attributed to a severe clinical entity known as "cytokine storm", which causes a rise in serum levels of many pro-inflammatory mediators and provokes massive tissue damage in several vital organs [7, 9].

In patients who developed severe symptoms, dyspnea was reported between 8–12 days after onset of symptoms, and some patients deteriorate into severe disease during the first week

after onset of symptoms. This accelerated worsening has been hypothesized to be caused by the cytokine storm and to thrombotic events that may be caused by infection with SARS-CoV-2 [10].

Hospitalized patients have been thoroughly described and analyzed, with an average time between onset of symptoms to intubation of 14.5 days, and a time from intubation to death ranging from 4–5 days [7, 9, 11]. A longer period between onset of symptoms and first contact seeking medical attention has been associated with a poorer outcome in these patients. However no in-depth studies have been conducted [12].

Until now, studies have been focused on patient-centered risk factors, while SARS-CoV-2 aggressiveness has been established as provoking 20% of severe and critic patients [13], however, there are still many unanswered questions concerning the clinical aggressiveness behavior of SARS-CoV-2. This study focuses on progression of symptoms as a marker of such aggressiveness, using the Period between initial symptoms and clinical progression to COVID-19 suspicion (PISYCS) to determine the risk of severe disease and mortality.

## Methods

### Study design and data source

A historic cohort study of Mexican patients that were classified as a suspect case of COVID-19 and sought medical attention in either public or private health services in Mexico, was analyzed. Data was provided by the General Directorate of Epidemiology of the Mexican Health Ministry, which is deidentified, publicly available online, and registers all patients in the Epidemiologic Surveillance System of the 32 federal states in Mexico. This analysis was done in all cases registered in this dataset up until April 25[th], 2020, with a total of 65,500 patients [14]. The Institutional Review Board of Anahuac University (Mexico City, Mexico) approved this study (Protocol approval #202044).

### Variable definition

Our dataset includes demographic characteristics such as age, gender, location, health sector, underlying medical conditions (obesity, diabetes, COPD, asthma, immunosuppression, hypertension, cardiovascular, chronic kidney diseases, and other comorbid diseases), pregnancy status, tobacco use, and indigenous language speaker. Furthermore, it references the dates of the onset of symptoms and the date of medical attention, including hospital admission, as well as the presence of pneumonia. Regarding in-hospital decisions, data include results of RT-PCR testing for SARS-CoV-2 (reported as positive, negative or pending), admission to the Intensive Care Unit (ICU) and requirement of mechanical ventilation. Finally, the date of death of all deceased patients is reported.

### Data analysis: Coding and substitution of variables

The state where the patient sought attention was recoded according to the socio-economic level of that particular state into low, middle and high level, based on the Gross Domestic Product (GDP) of that state, as reported by the National Institute of Statistics and Geography (INEGI) [15]. The health sector variable was recoded in three categories: private, with social security and without social security.

COVID-19 clinical suspicion was defined by the government health ministry´s official guidelines, as presenting with two of these symptoms: 1) cough, 2) fever or 3) headache, plus one or more of the following: 1) breathing difficulty, 2) sore or burning throat, 3) runny nose, 4) red eyes, 5) pain in muscles or joints, or 6) being part of these high-risk groups: pregnancy,

<5 or ≥60 years old, or having a chronic disease such as hypertension, diabetes mellitus, cancer or HIV.

These indications were broadcast on television, radio, newspaper and internet platforms since the beginning of the pandemic until July 1, 2020. Due to nationwide government issued stay-at-home orders, we assumed that medical attention was sought only when these criteria were met.

Upon medical evaluation, patients were asked to report the date of onset of initial symptoms (any combination of clinical features that appeared before meeting the previously mentioned criteria). Therefore, the Period between initial symptoms and clinical progression to COVID-19 suspicion (PISYCS) was established as the number of days between appearance of initial symptoms and the date in which the patients sought medical attention.

**PISYCS.** Initially, PISYCS was categorized in days (<1, 1, 2, 3, etc.), but to improve comprehension and data management, adjacent days whose frequency of death remained in similar proportions, were grouped into 4 categories (<24 hours, 1–3 days, 4–7 days and >7 days). The primary outcomes were mortality and pneumonia. The presence of pneumonia was used as an indicator of severe disease as reported in previous studies [7, 9].

Missing data were substituted using the mode for the following categorical variables: Health Sector (338 patients, 0.5%), indigenous language speaker (1241, 2%), tobacco use (242, 0.4%), pregnancy status (166, 0.3%), diabetes (255, 0.4%), COPD (245, 0.4%), asthma (251, 0.4%), immunosuppression (259, 0.4%), hypertension (241, 0.4%), cardiovascular disease (252, 0.4%), obesity (220, 0.3%), chronic kidney disease (245, 0.4%), other comorbid condition (331, 0.5%), admission to the ICU (12, <0.01%) and mechanical ventilation (12, <0.01%).

## Statistical analysis

Demographic features and comorbid conditions were initially compared between the four categories of PISYCS; age was analyzed with a one-way ANOVA and the rest of the variables with Chi squared test analysis. Afterwards, we performed a bivariate analysis comparing the four categories of PISYCS with the medical decisions made (result of PCR testing for SARS-CoV-2, hospital admission, ICU, mechanical ventilation) and outcomes (mortality and pneumonia) using the Chi squared test.

Finally, the four categories of PISYCS against primary outcomes -mortality and pneumonia-, were compared using a multivariable logistic regression model. The model was adjusted in five steps for the following variables: age, gender, indigenous language speaker, state's socioeconomic status, pregnancy, tobacco use, obesity, hypertension, diabetes, asthma, COPD, cardiovascular disease, chronic kidney disease, immunosuppression, other comorbid conditions. This model was repeated in four groups of patients within the sample, depending on their RT-PCR testing result for SARS-CoV-2:

a. All patients: Every patient in the dataset regardless of their test result

b. Positives: Only patients with a positive test result

c. Negatives: Only patients with a negative test result

d. Pending: Only patients with pending results of the test

The PISYCS used as reference in the regression models, was 1–3 days based in the lower rate of mortality, observed in the results of the bivariate analysis. Each logistic regression model is presented with the Odds Ratio (OR) and its respective 95% Confidence Interval (CI$_{95\%}$). Statistical significance was set at $p < 0.05$ and performed with SPSS version 25.0 (IBM). The full model for the group of all patients can be found in S1 and S2 Tables.

## Results

The study population included 65,500 patients. Among them, the average age was 41±17 years, 50.2%, were women, 55.8% belonged to a high socioeconomic level, 27.7% to a medium and 16.5% to a low one, 4.6% of patients were treated on a private health institution, 37.7% in a facility for patients with social security and 57.7% attended to a public hospital for patients without social security. Of all the patients, 41% had at least one comorbidity, hypertension being the most frequent in 17%, followed by obesity in 15.6% and diabetes 12.8%. In addition, 9.9% reported tobacco use and 2.3% of women were pregnant. Mortality was observed in 2176 patients (3.32%), and Pneumonia in 11568 patients (17.66%).

According to PISYCS patients were distributed as follows: 14.89% in <24 hours, 43.25% between 1–3 days, 31.87% between 4–7 days and 9.97% after 7 days, with no significant difference by gender. We compared PISYCS against demographic features and comorbidities. A PISYCS of <24 hours was more frequent in older patients (25.7% in patients > 80 years old vs. 15% in <30 years old) reversing in the period of 1–3 days (41.5% vs 48.4%, respectively), and returning to the initial behavior in subsequent periods. This same pattern was observed when comparing PISYCS with the presence of all comorbidities, except for asthma and obesity. Demographic Characteristics of all patients are summarized in Table 1.

Table 2 shows the initial medical decisions according to their PISYCS. More people were hospitalized in the first 24 hours (43.2%), with a drop towards the period of 1–3 days (19.7%), and a slight increase in subsequent days. A similar phenomenon is observed in terms of admission to the ICU, with admission being 2.8% when the period is <24 hours, falling to 1.8% in 1–3 days, and gradually increasing to 2.7% in 4–7 days and 3.2% in >7 days. The proportion of patients under mechanical ventilation steadily increased over time, starting from 1.6% in the period of <24 hours, up to 2.9% in the period of >7 days.

Table 3 and Fig 1 show the risks for mortality and pneumonia related to PISYCS. A "U-shaped distribution" was observed according to PISYCS (<24 hrs., followed by 1–3 days, 4–7, and >7). The proportion of patients with Mortality was 5.2%, 2.5%, 3.6%, and 4.1% (p<0.001), and for Pneumonia 22.5%, 14%, 19.5% and 20.6% (p<0.001). The adjusted-risk of mortality for all patients evaluated for clinical suspicion of COVID-19 according to PISYCS, was for <24 hours OR of 1.75 (95% CI, 1.55 to 1.98, p = <0.001), 1–3 days OR = 1 (reference value), 4–7 days, OR 1.53 (1.37–1.70, p = <0.001), and >7 days, OR 1.67 (1.44–1.94, p = <0.001), while for Pneumonia it was for <24 hours OR of 1.49 (95% CI, 1.39 to 1.58, p = <0.001), 1–3 days OR = 1 (reference value), 4–7 days, OR 1.48 (1.41–1.56, p = <0.001), and >7 days, OR 1.57 (1.46–1.69, p = <0.001).

## Discussion

In this study, we found an association concerning the Period between initial symptoms and clinical progression to COVID-19 suspicion (PISYCS) with the risk of severe disease and mortality in patients with suspected COVID-19. A "U shaped" distribution was observed, with a high risk of death and pneumonia when PISYCS is <24 hours (OR 1.75, and 1.49, respectively), with a decrease of this risk in 1–3 days (OR 1), and with an additional rise in subsequent periods of 4–7 days (OR 1.53, and 1.48) and >7 days (OR 1.67 and 1.57).

The increased risk of mortality and pneumonia observed in patients with PISYCS <24 hours, may be associated with the presence of a cytokine storm, which has been previously described as an early factor for severity [16]. This phenomenon is due to the uncontrolled release of pro-inflammatory mediators that lead to apoptosis of epithelial and endothelial lung cells, causing vascular extravasation, alveolar edema and hypoxia [17]. This inflammatory response in conjunction with the production of reactive oxygen species triggers an acute

**Table 1. Patient demographic characteristics according to period between initial symptoms and clinical progression to COVID-19 suspicion (PISYCS).**

| | | Period between initial symptoms and clinical progression to COVID-19 suspicion (PISYCS) | | | | | | | | Global P Value |
|---|---|---|---|---|---|---|---|---|---|---|
| | | <24 Hrs (n = 9759) | | 1–3 Days (n = 28331) | | 4–7 Days (n = 20877) | | >7 Days (n = 6533) | | |
| | | N | % | N | % | N | % | N | % | |
| Gender | Female | 4686 | 14.3%* | 14371 | 43.7% | 10562 | 32.1% | 3260 | 9.9% | <0.001 |
| | Male | 5073 | 15.6% | 13960 | 42.8% | 10315 | 31.6% | 3273 | 10.0% | |
| Age | < 30 | 2469 | 15.0%* | 7965 | 48.4% | 4732 | 28.7%** | 1302 | 7.9%*** | <0.001 |
| | 30–49 | 3745 | 12.8% | 12813 | 43.6% | 9785 | 33.3% | 3016 | 10.3% | |
| | 50–59 | 1430 | 14.7% | 3875 | 39.7% | 3321 | 34.1% | 1123 | 11.5% | |
| | 60–69 | 1095 | 19.8% | 2006 | 36.2% | 1785 | 32.2% | 656 | 11.8% | |
| | 70–79 | 615 | 21.9% | 1018 | 36.3% | 863 | 30.7% | 311 | 11.1% | |
| | > 80 | 405 | 25.7% | 654 | 41.5% | 391 | 24.8% | 125 | 7.9% | |
| | Mean (SD) | 43(20)* | | 40 (18) | | 42 (17)** | | 44 (16)*** | | <0.001 |
| State's Socioeconomic Status | High | 5532 | 15.1%* | 15745 | 43.1% | 11352 | 31.1%** | 3921 | 10.7%*** | <0.001 |
| | Medium | 2878 | 15.8% | 8027 | 44.2% | 5868 | 32.3% | 1402 | 7.7% | |
| | Low | 1349 | 12.5% | 4559 | 42.3% | 3657 | 33.9% | 1210 | 11.2% | |
| Health Sector | Private | 468 | 15.6%* | 1228 | 40.9% | 885 | 29.5%** | 418 | 13.9%*** | <0.001 |
| | With SS | 4584 | 18.6% | 10515 | 42.6% | 7274 | 29.5% | 2321 | 9.4% | |
| | Without SS | 4707 | 12.5% | 16588 | 43.9% | 12718 | 33.6% | 3794 | 10.0% | |
| Indigenous Language Speaker | Yes | 98 | 14.4% | 277 | 40.8% | 242 | 35.6% | 62 | 9.1% | 0.20 |
| | No | 9661 | 14.9% | 28054 | 43.4% | 20635 | 31.8% | 6471 | 10.0% | |
| Pregnancy | Yes | 139 | 18.3% | 383 | 50.5% | 190 | 25.1%** | 46 | 6.1%*** | <0.001 |
| | No | 9620 | 14.9% | 27948 | 43.2% | 20687 | 32.0% | 6487 | 10.0% | |
| Tobacco Use | Yes | 946 | 14.5% | 2786 | 42.8% | 2122 | 32.6% | 661 | 10.1% | 0.49 |
| | No | 8813 | 14.9% | 25545 | 43.3% | 18755 | 31.8% | 5872 | 10.0% | |
| Diabetes | Yes | 1514 | 18.1%* | 3307 | 39.5% | 2684 | 32.1%** | 867 | 10.4%*** | <0.001 |
| | No | 8245 | 14.4% | 25024 | 43.8% | 18193 | 31.8% | 5666 | 9.9% | |
| COPD | Yes | 401 | 22.6%* | 685 | 38.6% | 535 | 30.2% | 153 | 8.6% | <0.001 |
| | No | 9358 | 14.7% | 27646 | 43.4% | 20342 | 31.9% | 6380 | 10.0% | |
| Asthma | Yes | 394 | 12.4%* | 1422 | 44.6% | 1034 | 32.5% | 336 | 10.5% | 0.001 |
| | No | 9365 | 15.0% | 26909 | 43.2% | 19843 | 31.8% | 6197 | 9.9% | |
| Immunosuppression | Yes | 455 | 26.1%* | 697 | 40.0% | 429 | 24.6%** | 163 | 9.3% | <0.001 |
| | No | 9304 | 14.6% | 27634 | 43.3% | 20448 | 32.1% | 6370 | 10.0% | |
| Hypertension | Yes | 1932 | 17.4%* | 4415 | 39.7% | 3593 | 32.3%** | 1194 | 10.7%*** | <0.001 |
| | No | 7827 | 14.4% | 23916 | 44.0% | 17284 | 31.8% | 5339 | 9.8% | |
| Cardiovascular Disease | Yes | 470 | 22.0%* | 820 | 38.3% | 627 | 29.3% | 224 | 10.5%*** | <0.001 |
| | No | 9289 | 14.7% | 27511 | 43.4% | 20250 | 32.0% | 6309 | 10.0% | |
| Obesity | Yes | 1381 | 13.5% | 4093 | 40.0% | 3633 | 35.5%** | 1119 | 10.9%*** | <0.001 |
| | No | 8378 | 15.2% | 24238 | 43.9% | 17244 | 31.2% | 5414 | 9.8% | |
| Chronic Kidney Disease | Yes | 435 | 28.1%* | 612 | 39.5% | 374 | 24.1%** | 129 | 8.3% | <0.001 |
| | No | 9324 | 14.6% | 27719 | 43.3% | 20503 | 32.1% | 6404 | 10.0% | |
| Other Comorbid Condition | Yes | 662 | 19.0%* | 1440 | 41.3% | 1059 | 30.4% | 323 | 9.3% | <0.001 |
| | No | 9097 | 14.7% | 26891 | 43.4% | 19818 | 32.0% | 6210 | 10.0% | |

SS = Social Security. COPD = Chronic Obstructive Pulmonary Disease. SD = Standard Deviation. Statistical Significance p<0.05

* Significant Difference between periods of <24 hrs. and 1–3 days

** Significant Difference between periods of 4–7 days and 1–3 days.

*** Significant Difference between periods of >7 days and 1–3 days.

**Table 2.  Medical decisions according period between initial symptoms and clinical progression to COVID-19 suspicion (PISYCS).**

| | | Period between initial symptoms and clinical progression to COVID-19 suspicion (PISYCS)PISYCS | | | | | | | | Global P Value |
|---|---|---|---|---|---|---|---|---|---|---|
| | | <24 Hrs (n = 9759) | | 1–3 Days (n = 28331) | | 4–7 Days (n = 20877) | | >7 Days (n = 6533) | | |
| | | N | % | N | % | N | % | N | % | |
| **Type of care** | **Ambulatory Care** | 5547 | 56.8%* | 22742 | 80.3% | 15674 | 75.1%** | 4837 | 74.0%*** | <0.001 |
| | **Hospital Admission** | 4212 | 43.2% | 5589 | 19.7% | 5203 | 24.9% | 1696 | 26.0% | |
| **Admission to the ICU** | | 271 | 2.8%* | 503 | 1.8% | 563 | 2.7%** | 208 | 3.2%*** | <0.001 |
| **Mechanical Ventilation** | | 155 | 1.6% | 479 | 1.7% | 525 | 2.5%** | 189 | 2.9%*** | <0.001 |
| **Result of RT-PCR Test+** | **Not Positive to SARS-CoV 2** | 7036 | 81.4%* | 19801 | 80.4% | 12672 | 69.8%** | 3910 | 66.5%*** | <0.001 |
| | **Positive to SARS-CoV-2** | 1601 | 18.6% | 4818 | 19.6% | 5462 | 30.2% | 1961 | 33.5% | |

ICU = Intensive Care Unit. PCR = Polymerase Chain Reaction. SD = Standard Deviation.

Statistical Significance p<0.05

*Significant Difference between periods of <24 hrs and 1–3 days.

**Significant Difference between periods of 4–7 days and 1–3 days.

***Significant Difference between periods of >7 days and 1–3 days. + Undefined were not included.

respiratory distress syndrome (ARDS) leading to pulmonary fibrosis and death [18]. This could support the pharmacodynamic basis for the use of corticosteroids as adjuvant therapy in patients with COVID-19, which has been reported in other studies [19]. Chronic use of inhaled corticosteroids may be the reason why asthmatic patients manifest less severe symptoms [20], which was consistent with our results (See S1 and S2 Tables).

The increased risk of mortality and pneumonia in patients with PISYCS ≥4 days, could be explained by the thrombotic events that have been reported in patients with COVID-19. These events are caused by the excessive inflammation produced by the virus and platelet activation with accompanying endothelial damage [21, 22]. This occurs once the virus has colonized the respiratory system, impairing microvascular permeability, helping it spread even further. Hemostatic disorders are established by the presence of thrombocytopenia, and an increase in the D-dimer and fibrinogen, for which the use of antithrombotic therapies has been suggested [21, 23].

Additionally, an increased risk of lung superinfections must be considered. So far, bacterial and fungal pneumonias have been the most common etiologies. A study conducted in Wuhan, China reported a rate of lung superinfection from 5–27% in adults with COVID-19 [24]. Historically, superinfections have been associated with increased mortality in other viral respiratory infections, such as influenza [25].

Our findings may have further clinical implications if the pathophysiological processes were to be confirmed. The PISYCS could be useful as a prognostic marker and a decision-making tool for clinicians. Identifying individuals at higher risk of developing early-onset complications (with a PISYCS <24 hours) could justify a more aggressive treatment plan and monitorization strategies, focused on preventing complications of cytokine storm and ARDS. Additionally, patients with a higher risk of late-onset complications (with a PISYCS ≥4 days) could be identified and treated accordingly, justifying the use of thromboprophylaxis, preventing superinfections.

The PISYCS could also prove useful as a research categorization parameter for clinical studies exploring timing and efficacy of therapeutics. Immunomodulatory agents (such as IL-6 antagonists) and corticosteroids may only prove beneficial for patients with a PISYCS of < 24 hours and may further increase the risk of late-onset complications (superinfections) if used in a later PISYCS category [24].

**Table 3. Mortality and pneumonia among patients with COVID-19 according to the period between initial symptoms and clinical progression to COVID-19 suspicion (PISYCS).**

| Variable | Mortality | | | Pneumonia | | |
|---|---|---|---|---|---|---|
| **All Patients (N = 65,500)** | | | | | | |
| **PISYCS (N)** | **%*** | **OR (95% CI)** | **p-value** | **%*** | **OR (95% CI)** | **p-value** |
| < 24 Hours (9759) | 5.2% | 1.75 (1.55–1.98) | <0.001 | 22.5% | 1.49 (1.39–1.58) | <0.001 |
| 1–3 Days (28331) | 2.5% | 1 | 1 | 14% | 1 | 1 |
| 4–7 Days (20877) | 3.6% | 1.53 (1.37–1.70) | <0.001 | 19.5 | 1.48 (1.41–1.56) | <0.001 |
| > 7 Days (6533) | 4.1% | 1.67 (1.44–1.94) | <0.001 | 20.6% | 1.57 (1.46–1.69) | <0.001 |
| "p value" | <0.001 | | | <0.001 | | |
| *SUBGROUP ANALYSIS* | | | | | | |
| **Positive Test (N = 13,842)** | | | | | | |
| *PISYCS* | **OR (95% CI)** | | **p-value** | **OR (95% CI)** | | **p-value** |
| < 24 Hours | 2.11 (1.75–2.55) | | <0.001 | 1.66 (1.45–1.90) | | <0.001 |
| 1–3 Days | 1 | | 1 | 1 | | 1 |
| 4–7 Days | 1.42 (1.22–1.65) | | <0.001 | 1.69 (1.53–1.85) | | <0.001 |
| > 7 Days | 1.22 (1.00–1.49) | | 0.053 | 1.83 (1.62–2.07) | | <0.001 |
| **Negative Test (N = 43,419)** | | | | | | |
| *PISYCS* | **OR (95% CI)** | | **p-value** | **OR (95% CI)** | | **p-value** |
| < 24 Hours | 1.55 (1.29–1.86) | | <0.001 | 1.58 (1.46–1.70) | | <0.001 |
| 1–3 Days | 1 | | 1 | 1 | | 1 |
| 4–7 Days | 1.00 (0.83–1.21) | | 0.967 | 1.10 (1.01–1.16) | | 0.038 |
| > 7 Days | 1.39 (1.10–1.81) | | 0.014 | 1.11 (1.00–1.24) | | 0.061 |
| **Test Result Pending (N = 8,239)** | | | | | | |
| *PISYCS* | **OR (95% CI)** | | **p-value** | **OR (95% CI)** | | **p-value** |
| < 24 Hours | 1.41 (0.68–2.90) | | 0.357 | 0.80 (0.64–0.99) | | 0.043 |
| 1–3 Days | 1 | | 1 | 1 | | 1 |
| 4–7 Days | 1.22 (0.70–2.13) | | 0.484 | 2.03 (1.77–2.34) | | <0.001 |
| > 7 Days | 1.71 (0.78–3.74) | | 0.182 | 1.96 (1.57–2.44) | | <0.001 |

This Global Multiple Logistic Regression Model is adjusted for all demographic characteristics and comorbid conditions present in the patients. Adjustments for the group of all patients can be found in S1 and S2 Tables.

*Bivariate analysis between PISYCS vs. Mortality or Pneumonia.

Using a database fed by large numbers of people carries its risk, such as data inaccuracy. However, this imprecision is expected to be random and data are consistent with results of previous studies. Furthermore, we set April 25[th], 2020 as our cut-off date with the aim of including patients treated at an early stage of the pandemic in Mexico, at a time when hospitals were not yet working at overcapacity. This increases the probability of good quality of healthcare, decreases confounding factors for the outcomes evaluated because all required medical decisions could be made and were not limited by medical resources available at the time (i.e. number of ventilators or ICU beds).

Plenty of studies have described the incubation period and hospital stay of affected patients [7, 26, 27]. However, nobody has considered the progression of symptoms in patients with COVID-19 (PISYCS), as a guide for explaining the time-specific pathophysiology associated with the U-Shaped SARS-CoV-2 aggressiveness. Further studies are needed to corroborate the time-related pathophysiology behind these findings. Eventually, this could help identify specific therapies aimed towards the temporal progression of the disease.

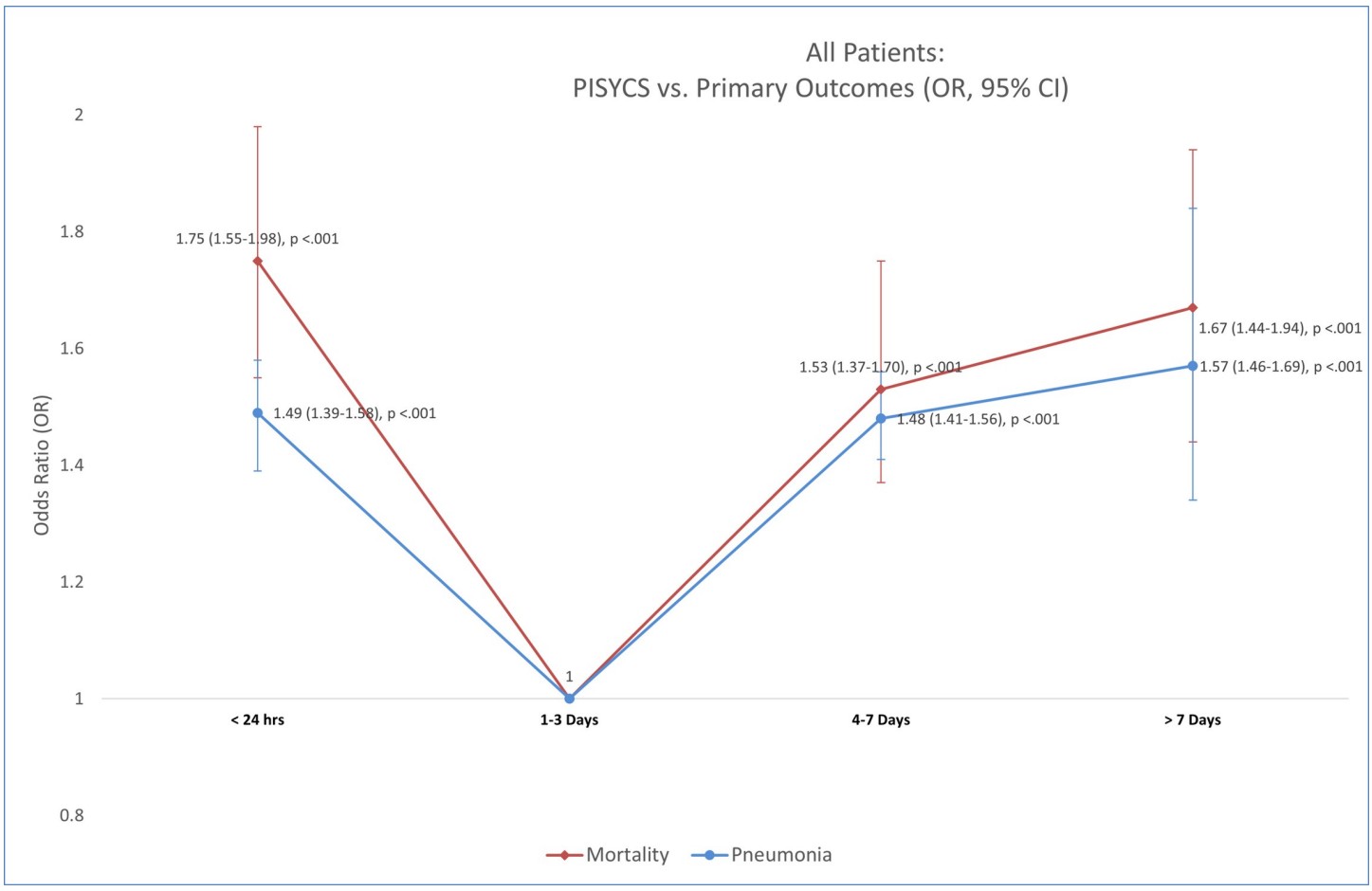

**Fig 1. U-shaped distribution of the odds ratio for the primary outcomes (mortality/pneumonia) vs. PISYCS.** Association according to the Period between initial symptoms and clinical progression to COVID-19 suspicion (PISYCS) and the primary outcomes of the study (Mortality and Pneumonia), including all patients. A U-shaped distribution is observed, with higher OR for PISYCS <24 hours and ≥4 days.

## Supporting information

**S1 Table. Stepwise analyses for binary logistic regression with mortality as outcome for all patients in the study.**
(XLSX)

**S2 Table. Stepwise analyses for binary logistic regression with pneumonia as outcome for all patients in the study.**
(XLSX)

## Author Contributions

**Conceptualization:** Dan Morgenstern-Kaplan, Bruno Buitano-Tang, Mercedes Martínez-Gil, Andrea Zaldívar-Pérez Pavón, Juan O. Talavera.

**Data curation:** Dan Morgenstern-Kaplan, Bruno Buitano-Tang, Juan O. Talavera.

**Formal analysis:** Dan Morgenstern-Kaplan, Bruno Buitano-Tang.

**Investigation:** Mercedes Martínez-Gil, Andrea Zaldívar-Pérez Pavón.

**Methodology:** Bruno Buitano-Tang, Mercedes Martínez-Gil, Andrea Zaldívar-Pérez Pavón, Juan O. Talavera.

**Software:** Dan Morgenstern-Kaplan, Bruno Buitano-Tang.

**Supervision:** Juan O. Talavera.

**Visualization:** Bruno Buitano-Tang, Andrea Zaldívar-Pérez Pavón.

**Writing – original draft:** Dan Morgenstern-Kaplan, Bruno Buitano-Tang, Mercedes Martínez-Gil, Andrea Zaldívar-Pérez Pavón.

**Writing – review & editing:** Dan Morgenstern-Kaplan, Bruno Buitano-Tang, Mercedes Martínez-Gil, Andrea Zaldívar-Pérez Pavón, Juan O. Talavera.

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
