## [Decision Letter · Decision Letter 0]

2 Nov 2020

PONE-D-20-23492

U-shaped-aggressiveness of SARS-CoV-2: Period between onset of nonspecific-specific symptoms for COVID-19. A population-based cohort study

PLOS ONE

Dear Dr. Talavera,

Thank you for submitting your manuscript to PLOS ONE. After careful consideration, we feel that it has merit but does not fully meet PLOS ONE’s publication criteria as it currently stands. Therefore, we invite you to submit a revised version of the manuscript that addresses the points raised during the review process.

Specific and no specific symptoms should be better described. The role of comorbidities is clinically interesting. We suggest to better describe it, elucidating, if possibile, the impact of each single comorbidities. We also suggest the Authors to describe the clinical impact of their findings. 

We look forward to receiving your revised manuscript.

Kind regards,

Chiara Lazzeri

Academic Editor

PLOS ONE

Journal Requirements:

Reviewers' comments:

Reviewer's Responses to Questions

**Comments to the Author**

1. Is the manuscript technically sound, and do the data support the conclusions?

Reviewer #1: Yes

2. Has the statistical analysis been performed appropriately and rigorously? 

Reviewer #1: Yes

3. Have the authors made all data underlying the findings in their manuscript fully available?

Reviewer #1: Yes

4. Is the manuscript presented in an intelligible fashion and written in standard English?

Reviewer #1: Yes

5. Review Comments to the Author

Reviewer #1: Dear Authors,

I consider your study original, interesting and well written. However, I have several suggestions to enhance the overall quality of the manuscript.

- Page 3, line 76; “SARS-CoV-2 aggressiveness has been stablished as provoking 20% of severe and critic patients” : please add a reference to this sentence.

- Please add in the methods text the ethic committee approval protocol number

- The distinction you propose about non-specific and specific symptoms lacks of a clear interpretation. All the proposed symptoms are specific of covid-19 disease. However, you described as “specific” the combination of conditions which allowed the medical/hospital admission. If so, the difference you proposed should be rediscussed or better explained. More than talk about “non-specific” and “specific” symptoms, It should be better to talk about “non-specific” and “specific” medical admission criteria. In this regard, the manuscript Title could should be revised too.

- When you discuss about covid-19 physiopathology, you could cite the review article of Dr. Pascarella et al. (COVID-19 diagnosis and management: a comprehensive review. J Intern Med. 2020 Aug;288(2):192-206. doi: 10.1111/joim.13091. Epub 2020 May 13. PMID: 32348588), which well resumes this mechanism.

- Among the PONSSs correlated with higher incidence of bad prognosis, it could be interest to discuss if there is a significant with any of the reported comorbidities (cardiovascular, diabetes, etc.). In this regard you could mention and include in this discussion and/or in the itroduction two recent observational studies published by Dr. Maddaloni et al: Clinical features of patients with type 2 diabetes with and without Covid-19: A case control study (CoViDiab I). Diabetes Res Clin Pract. 2020 Sep 21; PMID: 32971157; Cardiometabolic multimorbidity is associated with a worse Covid-19 prognosis than individual cardiometabolic risk factors: a multicentre retrospective study (CoViDiab II). Cardiovasc Diabetol. 2020 Oct 1; PMID: 33004045

- The results of your study show an inverse correlation between PONSS and clinical course severity. You may discuss the clinical relevance of this finding, proposing some management settings. For instance, a pharmacologic prophylaxis may be proposed from the onset of any COVID-19 symptom, even if non-specific for medical/hospital admission, especially in high risk patients, having a positive rt-PCR swab test.

Best Regards

6. PLOS authors have the option to publish the peer review history of their article (what does this mean?). If published, this will include your full peer review and any attached files.

Reviewer #1: No

---

## [Author Response · Author response to Decision Letter 0]

15 Nov 2020

The authors wish to thank the editors and reviewers for their time while reviewing this manuscript. We´ve addressed all comments and reviews in the manuscript, and the changes are described in the present document as well. 

Academic Editor Comments:

• Specific and no specific symptoms should be better described. 

o “Specific symptoms” and “non-specific symptoms” were replaced with “initial symptoms” and “COVID-19 suspicion”, respectively. These are further defined in the methods section of the article (page 5, lines 105-117)

• The role of comorbidities is clinically interesting. We suggest to better describe it, elucidating, if possible, the impact of each single comorbidities. 

o Although interesting, the role of comorbidities on COVID-19 severity has been thoroughly described in previous studies and the authors believe that is beyond the scope of this particular study, which focuses mainly on clinical manifestation of the disease and possible pathophysiological phenomena that could explain early and late onset mortality. 

o However, the full model analysis was adjusted for many comorbidities (diabetes, asthma, cardiovascular disease, etc.), which is described with further detail in the supplementary materials. 

• We also suggest the Authors to describe the clinical impact of their findings.

o Further explanation on the clinical impact of the use of PISCS was added to the discussion (page 12 lines 232-241). We believe that the PISCS could be useful as a prognostic marker to guide therapy once the pathophysiological processes have been better elucidated. 

Reviewer comments:

Reviewer #1

• I consider your study original, interesting and well written. However, I have several suggestions to enhance the overall quality of the manuscript

o The authors wish to thank reviewer 1 for the comments and reviews, the following changes have been added.

• Page 3, line 76; “SARS-CoV-2 aggressiveness has been stablished as provoking 20% of severe and critic patients”: please add a reference to this sentence.

o Thank you for the remark, a reference has been added to this sentence (citation #13 Azoulay et. al). 

• Please add in the methods text the ethic committee approval protocol number

o The ethics committee approval number is 202044, this has been added to the manuscript (page 4, line 89)

• The distinction you propose about non-specific and specific symptoms lacks of a clear interpretation. All the proposed symptoms are specific of covid-19 disease. However, you described as “specific” the combination of conditions which allowed the medical/hospital admission. If so, the difference you proposed should be rediscussed or better explained. More than talk about “non-specific” and “specific” symptoms, It should be better to talk about “non-specific” and “specific” medical admission criteria. In this regard, the manuscript Title could should be revised too. 

o Thank you for the remark. “Specific symptoms” and “non-specific symptoms” were replaced with “initial symptoms” and “COVID-19 clinical suspicion”, respectively. These are further defined in the methods section of the article (page 5, lines 105-117)

o The PONSS was further changed to address the new definition, elucidating the difference between the initial symptoms and the clinical suspicion of COVID-19. 

• When you discuss about covid-19 physiopathology, you could cite the review article of Dr. Pascarella et al. (COVID-19 diagnosis and management: a comprehensive review. J Intern Med. 2020 Aug;288(2):192-206. doi: 10.1111/joim.13091. Epub 2020 May 13. PMID: 32348588), which well resumes this mechanism.

o Thank you for the suggestion, this is a very comprehensive review and was added as a citation (page 11 line 216 citation #17)

• Among the PONSSs correlated with higher incidence of bad prognosis, it could be interest to discuss if there is a significant with any of the reported comorbidities (cardiovascular, diabetes, etc.). In this regard you could mention and include in this discussion and/or in the itroduction two recent observational studies published by Dr. Maddaloni et al: Clinical features of patients with type 2 diabetes with and without Covid-19: A case control study (CoViDiab I). Diabetes Res Clin Pract. 2020 Sep 21; PMID: 32971157; Cardiometabolic multimorbidity is associated with a worse Covid-19 prognosis than individual cardiometabolic risk factors: a multicentre retrospective study (CoViDiab II). Cardiovasc Diabetol. 2020 Oct 1; PMID: 33004045

o The interaction between the PISCS (Previously known as PONSS) with comorbidities such as diabetes is beyond the scope of this article, furthermore, the model was adjusted for these comorbidities and can be found in the supplementary material section. 

o Both papers about the interaction of COVID-19 with diabetes are of clinical importance and were cited in the introduction (citation #5-6)

• The results of your study show an inverse correlation between PONSS and clinical course severity. You may discuss the clinical relevance of this finding, proposing some management settings. For instance, a pharmacologic prophylaxis may be proposed from the onset of any COVID-19 symptom, even if non-specific for medical/hospital admission, especially in high-risk patients, having a positive rt-PCR swab test.

o We recognize that further details about the clinical impact of these findings were necessary, therefore they were added to the discussion section, in which we propose the use of PISCS as a prognostic marker to guide COVID-19 therapy once the pathophysiology of disease is corroborated. (page 12 lines 232-241)

---

## [Editor Report · Decision Letter 1]

19 Nov 2020

U-shaped-aggressiveness of SARS-CoV-2: Period between initial symptoms and clinical progression to COVID-19 suspicion. A population-based cohort study

PONE-D-20-23492R1

Dear Dr. Talavera,

We’re pleased to inform you that your manuscript has been judged scientifically suitable for publication and will be formally accepted for publication once it meets all outstanding technical requirements.

Kind regards,

Chiara Lazzeri

Academic Editor

PLOS ONE
---

## [Editor Report · Acceptance letter]

24 Nov 2020

PONE-D-20-23492R1 

U-shaped-aggressiveness of SARS-CoV-2: Period between initial symptoms and clinical progression to COVID-19 suspicion. A population-based cohort study 

Dear Dr. Talavera:

I'm pleased to inform you that your manuscript has been deemed suitable for publication in PLOS ONE. Congratulations! Your manuscript is now with our production department. 

Kind regards, 

on behalf of

Dr. Chiara Lazzeri 

Academic Editor

PLOS ONE